# Problems with Social Cognition and Decision-Making in Huntington’s Disease: Why Is it Important?

**DOI:** 10.3390/brainsci11070838

**Published:** 2021-06-24

**Authors:** Sarah L. Mason, Miriam Schaepers, Roger A. Barker

**Affiliations:** 1John van Geest Centre for Brain Repair, Department of Clinical Neurosciences, University of Cambridge, Cambridge CB2 0PY, UK; ms2709@cam.ac.uk (M.S.); rab46@cam.ac.uk (R.A.B.); 2NIHR Biomedical Research Centre, Addenbrookes Hospital, Cambridge CB2 0QQ, UK; 3Wellcome Trust Medical Research Council-Cambridge Stem Cell Institute, Cambridge CB2 0AW, UK

**Keywords:** social cognition, Huntington’s disease, decision-making

## Abstract

Huntington’s disease starts slowly and progresses over a 15–20 year period. Motor changes begin subtly, often going unnoticed by patients although they are typically visible to those close to them. At this point, it is the early non-motor problems of HD that arguably cause the most functional impairment. Approximately 65% of gene carriers will experience a reduction in their occupational level, and just under half will feel unable to manage their finances independently before a clinical diagnosis is made. Understanding what drives this impairment in activities of daily living is the key to helping people with HD to live more independently for longer, especially in early disease. Early cognitive decline is likely to play a contributory factor although few studies have looked directly at this relationship. Recently, it has been shown that along with the well documented dysexecutive syndrome seen in HD, changes in social cognition and decision-making are more common than previously thought. Furthermore, some of the early neuropathological and neurochemical changes seen in HD disrupt networks known to be involved in social functioning. In this review, we explore how HD changes the way individuals interact in a social world. Specifically, we summarise the literature on both classical and social decision-making (value-based decision-making in a social context) along with studies of theory of mind, empathy, alexithymia, and emotion recognition in HD. The literature specific to HD is discussed and supported by evidence from similar neurodegenerative disorders and healthy individuals to propose future directions and potential therapeutic avenues to be explored.

## 1. Introduction

Changes in social function are a prominent feature of many neurodegenerative disorders, but it is becoming increasingly apparent that this is especially true in patients with Huntington’s disease (HD). HD is an autosomal dominant neurodegenerative disease that progresses over a 15–20 year period [1] and typically presents with a combination of cognitive, motor, and neuropsychiatric problems. Motorically, choreic movements, unsteady gait, and dysarthria are common. This is accompanied by widespread changes in cognitive function including abnormalities of psychomotor speed, working memory, emotion recognition, and executive functions [2,3]. Finally psychiatric symptoms are common but vary widely in their severity. Despite the motor features being the most visible signs of HD, patients and their families often identify the non-motor symptoms as the most debilitating [4].

Until recently, the cognitive impairments in HD were thought to be exclusively a result of damage to the frontostriatal networks and, consequently, little attention was paid to cognitive domains other than executive function. Recently, however, there has been increasing acknowledgement that wider neural networks are affected by the disease process in HD, including structures within the limbic system [5,6]. The limbic system is a functional concept that links discrete brain regions including the amygdala, hippocampus, hypothalamus, and thalamus due to their overlapping involvement in processes, such as emotion response, sleep modulation, motivational behaviour, memory, and social cognition. All of which are affected in HD.

In this review, we will focus on the emerging evidence of impairments in social cognition and social decision-making in HD. We will discuss the literature demonstrating that there are impairments in the way HD patients gather social information and, how this leads to problems using that social information to make decisions in everyday life. Finally, in the absence of any current therapeutic options, we will postulate about potential approaches to treatments that could be explored in the future for these types of issues.

## 2. Social Impairment in HD

Anecdotal evidence from clinical practice highlights the diverse range of social problems that HD patients can experience. They might be insensitive, tactless, and distrusting towards those closest to them whilst disproportionately entrusting of strangers, thereby alienating themselves from their support networks, leaving them vulnerable to manipulation. Disinhibited behaviour can leave patients (and their families) ostracised from friendship groups and increasing social withdrawal often results in feelings of isolation and loneliness. A lack of awareness of HD by the general public and a poor understanding of its clinical features, can lead to patients being erroneously described as appearing “drunk”, especially in the early stages of the condition when other more obvious features are not present. Possibly leading to further exclusion affecting the patient’s independence.

The framework that underpins successful social living is complex, involving multiple interdependent processes. Simplistically, it requires people to possess the ability to: (a) detect and process social information, (b) understand their own mental states and those of others around them (mentalize), (c) bond and form social relationships with other people, and (d) make adaptive social decisions. Behaviour needs to be flexible and able to respond dynamically to a range of diverse social situations but can also be influenced by socio-economic features, cognitive capacity (attention, working memory, sensory processing), and changes in social cognition.

Social cognition refers to the processes that underlie the way in which people interact in a social world. In reality, it deals with the ways in which an individual gathers social information from the environment around them and makes sense of it in their own mind. It covers factors such as: emotion recognition, awareness (alexithymia) and reactivity, theory of mind (ToM) and empathy (Figure 1, Table 1).

In HD, social cognitive changes have been linked to levels of self-reported social distress [7] and are likely to underpin the observed social impairments.

To support the discussion, a comprehensive search of studies between January 1990 and January 2020 was conducted using the PUBMED database. The search terms used were: “Huntington’s disease” (TITLE) AND “Alexithymia” (TITLE/ABSTRACT) or “theory of mind” (TITLE/ABSTRACT), “empathy” (TITLE/ABSTRACT), “social decision-making” (TITLE/ABSTRACT). Studies were excluded if they were review articles, case-studies, not written in English, primarily focused on task validation or intended to establish neural/electrophysical correlates. A preferred reporting items for systemic reviews and meta-analyses (PRISMA) diagram providing details of the number of studies which met the search criteria is included in Figure 2.

## 3. How HD Changes Social Cognition

### 3.1. Emotional Recognition

Deficits in emotional recognition have been widely reported in pre-manifest and early HD (for review see [8]). It was initially proposed that disgust processing was preferentially targeted [9,10,11,12,13,14] with abnormalities found in multiple domains including disgust recognition in vocal [13], olfactory, and gustatory modalities [15,16], as well as on facial emotion recognition tasks [13,17,18]. However, not all studies have been able to replicate these findings [17,18,19,20,21,22,23,24]. A recent systematic review of the literature found that a deficit in anger recognition was the most consistently reported impairment in studies of manifest HD patients, closely followed by impairments in disgust, then fear recognition with problems of sadness and surprise recognition reported less frequently [8].

Calder et al. [25] looked in depth at disgust recognition in HD and found evidence to suggest that disgust and anger were not entirely dissociable. Patients were shown a series of different scenarios each containing one of 3 distinct disgust facial expressions: disgust associated with an unpleasant smell (nose wrinkle), disgust associated with an unpleasant taste (mouth gape), and disgust relating to violations of social or moral standards and interpersonal skills (lip curl). The authors suggested that situations that evoke this later disgust are often accompanied by feelings of anger as well as disgust, for example, we experience both disgust and anger when we hear reports of child abuse. Calder and colleagues showed that HD patients were significantly worse than controls at recognising the facial expression linked with moral disgust (lip curl) but showed only a borderline (insignificant) difference with both the other conditions.

In contrast, the literature in pre-manifest HD lacks any real consistency except to say that deficits are typically only reported in the recognition of negative emotions [8]. Although, it is worth noting that happiness is the only positive emotion included in these studies. Some healthy aging research has suggested that the positivity effect might be caused by ceiling effects (that happiness is a lot easier to distinguish from the negative emotions) [26]. Despite this, Johnson et al. [27] reported impaired recognition of anger, disgust, fear, sadness, and surprise in a cross-sectional population of 475 pre-manifest HD patients. Greater motor signs (although still below the threshold for diagnosis) were associated with less accurate recognition of all negative emotions and impaired recognition was associated with predicted time to onset. However, other studies have shown more focal deficits preferentially affecting one or more emotion, typically disgust or anger [9,11,19,28], or no abnormality at all [29]. Importantly, two recent studies have shown that performance on negative emotion recognition tasks can be predictive of impending clinical diagnosis [28,30].

Despite clear evidence that HD gene carriers have difficulty recognising emotions from faces, even before the onset of overt disease, the implications of this are unknown. Preliminary work has associated emotion recognition with emotional regulation in a very small group of manifest HD patients (n = 13) [31] and, in an even smaller sub-group of patients (n = 10) impaired emotion recognition has been shown to correlate with impairments in the perception of trustworthiness [32]. However, further work needs to be done to replicate and expand on these findings.

Studies in healthy individuals have shown that observing facial expressions strengthens decisions about whether to trust someone or not [33]. In alcoholism, difficulties with emotion recognition have been linked to interpersonal problems [34] and in Parkinson’s disease emotion recognition problems are associated with difficulties with social relationships [35] and with social behaviour disorders [36]. No such work has been undertaken to date in HD to our knowledge, and the extent to which problems with emotion recognition relate to functional disability and the problems with social interactions described above, remains unclear.

### 3.2. Alexithymia

Alexithymia is defined as a difficulty identifying, describing and regulating one’s emotions [37] resulting in problems with emotional awareness, communication, and insight (Table 1). Commonly measured using self-report questionnaires, such as the Toronto Alexithymia Scale (TAS20 or TAS26) [38], alexithymia is associated with emotion recognition problems [39,40] in healthy individuals, and has been linked to activation of the limbic system [41] and anterior insula [42] in functional imaging studies.

It is somewhat surprising, therefore, given the plethora of work describing emotion recognition problems in HD, that few have looked at the role of alexithymia. Two small studies in patients with mild HD, found that levels of alexithymia were equivalent to those found in matched controls using an adapted version of the TAS20 [43,44] despite showing a marked dampening of EEG activation during the voluntary production of emotional facial expression. Conversely, in a study of a larger group of 40 patients that used the TAS26 both manifest and premanifest gene carriers showed higher levels of alexithymia than matched controls [45].

Evidence from other neurological conditions have found that higher levels of alexithymia are associated with lower levels of empathy [46] and greater problems understanding others emotions or assuming their point of view [47]. In patients with TBI, it has also been associated with a worse social and family life [48] and was found to be a significant predictor for psychological distress [49] which the authors suggest was due to the development of maladaptive coping strategies.

### 3.3. Theory of Mind

The term ToM has been used to describe an individual’s ability to understand the presence of beliefs, feelings, intentions, and interests in other people that can differ from their own and from reality [50] (Table 1). Multiple brain regions have been linked with ToM, including the prefrontal cortex and superior temporal sulcus [51], although there is some variability depending upon the tasks used.

There is increasing evidence that ToM is impaired in HD, on both first order (reflecting on another’s thoughts and feelings) and second order (predicting another’s thoughts and feelings) ToM tasks [52]. Patients have been shown to have difficulty with understanding sarcasm [53,54] and ToM deficit were found on multiple different tasks including the faux pas task, the Reading the Mind in the Eyes Task [55] and cartoon based tasks [56,57]. Recent work has suggested that poorer ToM in HD is associated with a reduction in “altruistic punishment” in the Dictator Game [58]. Patients are less likely than controls to punish a third party for perceived unfair behaviour (e.g., refusing the proposed division of funds which results in both parties leaving with no money). It is unclear whether this is because patients have a different appreciation of what is “fair” or whether they are simply more self-interested and, therefore, less willing to lose their share of the money.

In most cases, ToM is impaired prior to onset of clinical disease [5,59] and gets [60] worse with disease progression [5]. A recent study suggested that the amount of information HD gene carriers gathered when visually scanning facial stimuli [61], in part, predicted their ability to accurately identify the thoughts, feelings, or emotions depicted in the stimuli. Furthermore, in manifest patients, while performance does not relate to levels of global cognitive functioning, it does appear to relate to visuospatial processing ability [62]. Further work is needed to determine whether or not ToM deficits are secondary to attentive problems or if patients have a fundamental problem mentalizing.

An increasing number of studies have found impaired ToM across multiple different cohorts of HD patients. Some have tried to establish the cause of these deficits, but few have looked to understand the consequences they bring. Although problems with ToM are assumed to play a role in the interpersonal difficulties experienced by patients, this has not been confirmed empirically. In the wider literature, having a better ToM has been associated with having a more varied social network [63] while lower competency on ToM tasks has been found to be predictive of self-reported depression and loneliness [64]. Importantly for the HD community, healthy individuals with a better ToM are also better able to detect deception in video vignettes [65]. Being able to identify if someone is trying to mislead you is essential in today’s world where the number of telephone, internet, and doorstep scams are growing daily. Better understanding the relationship between ToM, trust and deception in HD may provide important information to help safeguard patients.

### 3.4. Empathy

Empathy is the ability to understand and respond appropriately to another’s thoughts, feelings, and emotions by taking their perspective or “putting yourself in their shoes” [66]. It is an essential part of social interaction, although it has received very little attention in the HD literature. Responding empathetically to those around you is a necessary emotional competency that is required for building and maintaining social bonds [67].

The role of empathy in HD has been investigated by a few, small studies with mixed results. Generally, levels of empathy in premanifest [68] and manifest HD [44,52] were found to be similar to controls. However, where deficits have been found, they tend to be subtle and [69] related to specific problems with cognitive empathy (the ability to see the world from another person’s viewpoint) which are accompanied by problems with social-skills [68] and increased levels of personal distress [59].

Far more work needs to be done before any conclusions can be drawn about the role of empathy in the social impairment experienced in HD. Given the established links between empathy and social interaction and the emerging evidence that lower levels of empathy are associated with greater loneliness in older adults [70], this is an area worthy of further investigation.

**Table 1 brainsci-11-00838-t001:** Social cognition: processes relevant to Huntington’s disease.

	Processes	Assessment Scales	Neural Correlates
**Emotion** The subjective experience of physiological arousal in response to an event. Examples include happiness, sadness, surprise, fear, anger, disgust and contempt	**Recognition:** The ability to identify emotions in another person.	Ekman faces [71]	Amygdala, insula, globus pallidus, lateral orbitofrontal cortex [72]
	**Awareness (Alexithymia):** The ability to identify and describe emotion in oneself.	Toronto Alexithymia Scale [38] Perth Alexithymia Questionnaire [73]	Amygdala, dorsomedial prefrontal cortext, insula, precuneus, dorsal anterior cingulate [39]
	**Reactivity:** The ease at which someone become emotionally aroused and the intensity of emotional experiences	Emotion Reactivity Scale [74] Emotion Intensity Scale [75]	Amygdala, ventromedial prefrontal cortex [76]
**Theory of Mind** The ability to attribute mental states e.g., beliefs, intents, desires and emotions, to oneself an others Examples include: panicked, playful, jealous, excited	**Mentalising:** The ability to interpret the mental state of oneself and other.	Reading the Mind in the Eyes [77] Faux Pas [78] Happe-Frith animations [79]	Bilateral temporparietal junction, medial prefrontal cortex, right superior temporal sulcus [80,81]
	**Empathy:** The ability to feel what other people are feeling.	Empathy Quotient [82] Multidimensional Emotional Empathy Scale [83] Toronto Empathy Questionnaire [84]	Temporoparietal junction [85], medial prefrontal cortex [86]

## 4. How HD Changes Decision Making and Social Decision-Making

Decision-making (DM) involves the probabilistic evaluation of multiple alternatives, ideally leading to the best option being chosen [87]. It relies on a complex interplay of different cognitive processes to allow for the flexibility needed to adjust to a diverse range of situations [88]. Most everyday decisions are made within a social context. Individuals use social norms to guide their choices, they must consider the impact of their decisions on others and commonly engage in shared decision making with other people. DM in the real world is, therefore, heavily reliant on social cognitive functions [89]. Understanding how DM capacity changes with disease progression in HD is crucial. Patients must make many difficult but highly important decisions throughout the course of their disease, many of which have implications for family and friends around them (Table 2).

DM research in HD is yet to integrate the changes in social cognition detailed above but instead has been built around a DM model defining two types of DM processes, DM under risk and DM under ambiguity [90]. DM under risk is when all possible outcomes (including the probability distribution of each of these outcomes) are known, allowing for an informed evaluation and comparison of the options available. Whereas DM under ambiguity describes the situation where the different outcomes and the probability distribution of each of these outcomes is unknown. These two concepts capture differences in attitude towards uncertainty [90]. Gambling tasks including the Iowa Gambling task, Cambridge Gambling task, and the Game of Dice task have all been used to investigate this DM paradigm in HD (White, in prep).

However, contradictory results have been reported for studies comparing DM under risk and under ambiguity. Focusing on early manifest patients, Adjeroud et al. (2017) [91] found a reduced ability with DM under ambiguity but no differences were seen between patients and healthy controls in their DM ability under risk. Campbell et al. (2004) [92] came to the opposite conclusion suggesting there was an impairment in DM under risk but no group differences in DM under ambiguity. Finally, Holl et al. (2013) [93] found no group differences for both types of DM processes.

The remaining literature has focused exclusively on DM under risk presenting equally mixed results. Stout et al. (2001) [94] and Galvez et al. (2017) [95] were able to find significant group differences proposing impaired DM under risk in manifest HD patients. In contrast, Watkins et al. (2000) [96] and Minati et al. (2011) [97] found no differences in DM under risk between HD patients and healthy controls.

Few studies have focused on DM in pre-manifest gene expansion carriers, however the existing research at this point suggests there are no differences from healthy controls in DM under ambiguity or risk [91,98].

Differences in methodology and participant’s characteristics might have contributed to these conflicting results (White, in prep). However, it could also be argued that these basic gambling tasks are not sufficient to detect the self-reported DM deficits in HD. One of the defining factors of naturalistic DM is its complexity regarding available information and the cognitive skills required [88]. These highly controlled but simplistic tasks potentially reduce the level of cognitive demands to a degree that they become manageable for people who otherwise struggle in the real world [88]. In case of DM under risk, it is rarely the case that all outcomes and their probability distributions are known. For example, when deciding to speed while driving it is impossible to know how probable a crash might be, since many highly dynamic factors influence this possible outcome. DM under risk as defined by research might therefore be relatively rare in everyday life.

Assessments using only DM tasks are not sufficient when characterizing the DM abilities of an individual, it is also necessary to identify changes in their underlying cognitive functions as these may be contributing to the DM impairment. In HD there are many reported cognitive problems that are relevant to everyday DM, such as reward and punishment processing, social cognition, memory, executive functions, and motivation [1,2]. However, the gambling tasks used in HD research only challenge a small subset of those functions (e.g., executive functions, reward and punishment processing, memory). Further, only two DM studies included more detailed cognitive assessments using traditional neuropsychological tasks [91,94]. Both found that decreased performance on some measures of executive functions relate to maladaptive DM [91,94]. This is consistent with research in other neurodegenerative diseases proposing a strong relationship between executive function and basic DM [99]. Stout at al. (2001) [94] further found that HD participants who performed better on the DM task also had better memory abilities. The proposed explanation was that reduced memory might impair the patient’s ability to learn from reward and punishment, leading to an inability to adjust their behaviour. It is probable that reduced executive functions and memory underlie the reported functional DM deficit in HD, as these represent the most fundamental cognitive functions necessary for DM actions [99]. However, it is unlikely that they exclusively cause the clinical DM impairment, as everyday DM does involve a much larger set of cognitive functions [100]. Many of those functions are commonly found to be altered in HD, including social cognition [101], reward and punishment processing [102], and metacognition [103]. Therefore, future research on DM in HD will require a more comprehensive battery of cognitive tests, which could be achieved by focusing on specific everyday DM scenarios and not social DM in general.

The issue of over simplistic DM task not providing valuable information on real world DM has been recognised by the field of economics and naturalistic DM, which has led to the development of more ecologically valid tasks [87,88]. Specifically, neuroeconomic games have been a valuable resource to assess the impact that social context might have on DM [87]. These paradigms mimic DM in a social environment consisting of game play situations with multiple participants involved, creating either a competitive or collaborative environment [104]. A large body of research focusing on psychiatric disorders associated with social deficits has successfully employed these neuroeconomic games, recognising their potential as diagnostic tools for such deficits [104,105]. More recently research on vulnerability towards financial exploitation in healthy older adults and older adults with dementia have also included these assessments, which has led some to propose that such approaches may be more appropriate in acquiring clinically relevant information on susceptibility to financial abuse and financial DM capacity [106,107,108,109,110]. Consequentially, as HD is associated with a deficit in social cognition these neuroeconomic games might be a more ecologically valid approach to assess DM in HD, as they include social components creating a more realistic environment compared to traditional DM tasks. Therefore, a first step to investigate for a functional DM deficit in HD may be to assess social DM, targeting a DM situation likely to be affected in HD.

The above underlines the need for more research to bridge the gap between laboratory-based DM research and real-world functioning in patients and age matched controls, as the existing research in HD has exclusively focused on very simplistic models of DM without considering it in the context of everyday functioning. This will help to improve the assessment of DM capacity, identify vulnerable individuals, and develop more individualised care for such people.

## 5. Potential Treatment Approaches

As with many aspects of HD there are currently no treatments available to help manage the social impairment in HD. Psychoeducation can help prepare patients and their families about the changes that may occur with the condition, as well as provide an explanation for some of their challenging behaviours as and when they arise. However, while psychosocial interventions, such as cognitive stimulation theory, are commonplace in other dementias [111], their efficacy has never been formally tested in HD.

Looking to the wider literature, behavioural therapies for ToM, empathy emotion recognition and alexithymia, are beginning to be used with some degree of success although this work is still in its infancy. To date, the only approach used directly with an HD population has been a self-guided online training program which evoked a significant improvement in emotion recognition accuracy in a small group (N = 22) of premanifest and early HD participants [112].

Anecdotally, alexithymia may be responsive to cognitive behavioural therapy (CBT), group therapy [113] or psychotherapy although the empirical evidence supporting this is sparce. Focused training strategies using neurocognitive training [114] and imitation therapy [115] have be found to improve ToM in patients with schizophrenia. Similarly, motivational-based interventions have been shown to boost empathy in healthy college students with real-life implications including an increase in their number of friends [116]. These methodologies appear to be accessible for an early or premanifest HD population, which is also the group who would benefit most, therefore these strategies may be a future therapeutic avenue worth exploring.

Going forward, a better understanding of the underlying neural basis for these deficits in social cognition in HD would be helpful. Some brain structures (e.g., amygdala) and neurotransmitters (e.g., oxytocin) have been heavily linked to the processing of social information and are thus obvious treatment targets.

In this respect, there is a 45% reduction in the number of oxytocin-expressing neurones and a 24% reduction in the vasopressin-expressing neurones in the HD hypothalamus [117] compared to controls even at very early stages of the disease [118]. Evidence from animal models suggests that an imbalance in the oxytocin-vasopressin networks may be linked to the psychiatric phenotype in HD [119]. In humans, oxytocin levels have been shown to correlate with performance on social cognition tasks in a very small (n = 12) population of early manifest HD patients [120]. Furthermore, intranasal administration normalised brain activity in response to disgust stimuli in 9 HD patients [121]. In healthy individuals, higher levels of oxytocin in the body have been shown to reduce anxiety and stress in social situations [122,123]. When manipulated artificially, oxytocin can increase trust [124], improve mentalizing performance on ToM tasks [125] and enhance amygdala dependent social learning and emotional empathy in healthy controls [126].

Taken together, there is sufficient justification to support further investigation of this pathway as a potential treatment for social impairment in HD.

## 6. Conclusions

Being able to function in a social world is an essential part of everyday life. Human beings are interdependent, social creatures who strive for a sense of “belonging” within a wider social group [127]. A sense of isolation or a perception of being disconnected from the environment around us leads to feelings of loneliness [128] which, in turn, has been linked to depression, anxiety, low life satisfaction, and suicidal ideation [129,130]. Social isolation has been shown to be a significant risk factor for dementia in later life [131] and conversely, larger and more complex social networks have been found to be associated with better cognitive function in patients with Alzheimer’s disease, independent of their pathology [132].

In other conditions, such as schizophrenia, problems with social cognition have been linked to poorer social outcomes and social functioning [133,134,135]. Social dysfunction and social isolation have a reciprocal relationship, whereby social dysfunction can lead to social isolation which, in turn, exacerbates the problem creating a positive feedback situation.

Research investigating social cognition in HD is still in its infancy. To date the work has lacked focus. Multiple studies have used small cohorts of heterogeneous patients to look at isolated aspects of social cognition and related performance back to stages of disease progression in a descriptive fashion. Little attention has been given to understanding how changes in social cognition relate to one another or to the functional impairment experienced by patients. Furthermore, symptomatic treatments are desperately needed for HD and, arguably, more attention should be directed on processes, such as social cognition given their relationship with quality of life and the impact on the patient’s ability to live independently. The foundations have been set but there is still a significant amount of work to do in this area over the next few years.

## Figures and Tables

**Figure 1 brainsci-11-00838-f001:**
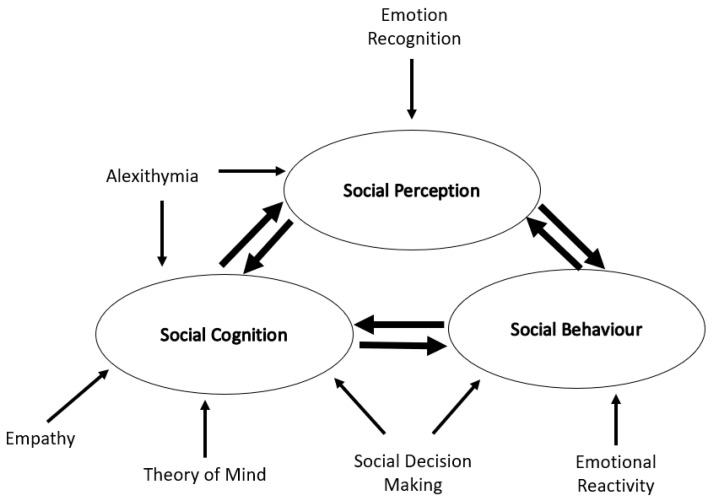
Social cognition in Huntington’s disease. A schematic representation of the social cognitive processes involved in Huntington’s disease and their relationship to one another.

**Figure 2 brainsci-11-00838-f002:**
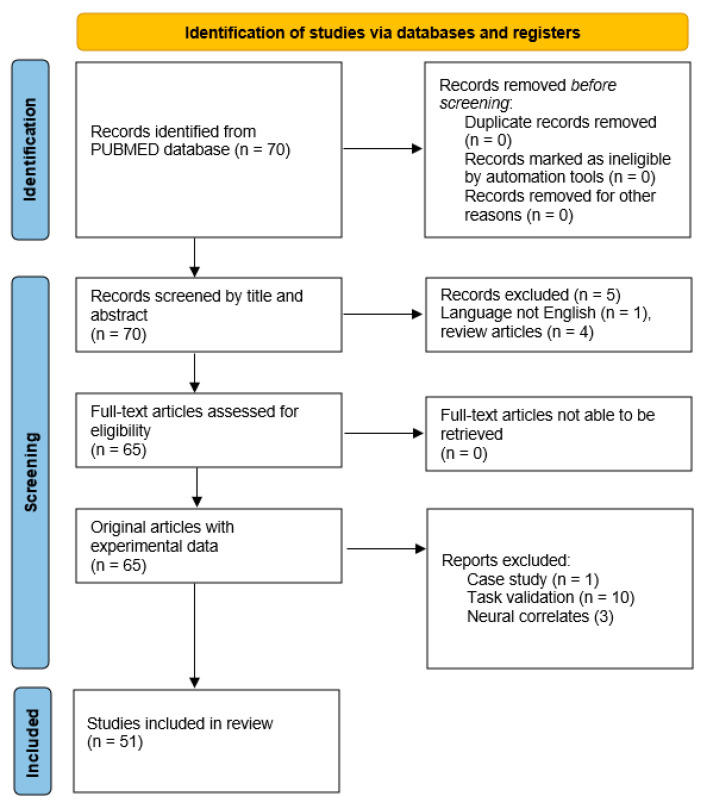
Search strategy: flowchart showing the number of studies included in the systemic review.

**Table 2 brainsci-11-00838-t002:** Examples of decisions Huntington’s disease gene carriers make.

Genetic testing	Whether or not to have predictive testing When to have predictive testing Whether to share the results of the test with family, friends and the wider world
Reproduction	Whether or not to have children at all Whether to have children who are also at risk of HD Whether to use preimplantation testing or prenatal testing or natural conception
Sharing HD status	When to tell friend or new partners about gene status Do employers need to know? When do the DVLA need to know? Does the GP need to know? When to tell children that HD is in the family
Forward planning	When or if to set up an advanced directive Is it necessary to set up a proxy decision-maker in advance? If so, who will it be?
Care choices	Do you want to have a PEG fitted to assist feeding, if so when should it be removed? When is independent living no longer practical or possible? Should care be provided at home or in a nursing care facility.
Experimental research	Whether to take part in experimental studies

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
