# Peer review of "Problems with Social Cognition and Decision-Making in Huntington’s Disease: Why Is it Important?"

_brainsci, 2021, doi:10.3390/brainsci11070838_

Round 1
Reviewer 1 Report
In what the manuscript content is concerned, the paper brings interesting and novel information about the impact of Huntington Disease on social cognition. Given the abundant results, presented in multiple distinct paragraphs (covering emotional recognition, alexithymia, theory of mind, empathy, decision-making and social decision-making), the final section about potential therapeutic approaches could be enriched, so that it addresses the above mentioned variables.
In terms of format, the manuscript is quite long and thematically-extensive to be considered a commentary, being closer to a review. However, to qualify as a review, it should provide critical details about the design, the procedure of studies selection etc. Each of these two possibilities implies a major revision of the manuscript body, either to make it more focused, either to offer the missing information.
Author Response
We thank the reviewer for their comments and for taking the time to read our manuscript.
Point 1: The final section about potential therapeutic approaches could be enriched, so that it addresses the above mentioned variables.
Response 1: In response to the first point, we agree that the final section on potential therapeutic approaches should be expanded. We have added the following two paragraphs and believe that this makes the section more rounded and comprehensive as a result:
"Looking to the wider literature, behavioural therapies for ToM, empathy emotion recognition and alexithymia are beginning to be used with some degree of success although this work is still in its infancy. To date, the only approach used directly with an HD population has been a self-guided online training program which evoked a significant improvement in emotion recognition accuracy in a small group (N=22) of premanifest and early HD participants [1].
Anecdotally, alexithymia may be responsive to cognitive behavioural therapy (CBT), group therapy [2] or psychotherapy although the empirical evidence supporting this is sparce. However, focused training strategies using neurocognitive training [3] and imitation therapy [4] have be found to improve ToM in patients with schizophrenia. Similarly, motivational-based interventions have been shown to boost empathy in healthy college students with real-life implications including an increase in their number of friends [5]. These methodologies appear to be accessible for an early or premanifest HD population, which is also the group who would benefit most, therefore these strategies may be a future therapeutic avenue worth exploring."
Point 2: In terms of format, the manuscript is quite long and thematically-extensive to be considered a commentary, being closer to a review. However, to qualify as a review, it should provide critical details about the design, the procedure of studies selection etc. Each of these two possibilities implies a major revision of the manuscript body, either to make it more focused, either to offer the missing information.
Response 2: In response to the second point, we fully agree that the manuscript as it is does not sit comfortably as either a review or a commentary. We believe that what makes this manuscript novel and of value to the HD community is the breadth of areas that it covers. Therefore we have taken your suggestions to adapt it and make it a review article. We have added the following information about the study selection to the text and included a further figure containing the PRISMA diagram for study selection. To ensure that this is an accurate reflection of the studies included with have double checked that all studies identified in the search are references in the manuscript and any that were not missing have now been included.
To support the discussion, an comprehensive search of studies bewteen January 1990 and January 2020 was conducted using the PUBMED database. The search terms used were: “Huntington’s disease” [TITLE] AND “Alexithymia” [TITLE/ABSTRACT] or “theory of mind” [TITLE/ABSTRACT], “empathy” [TITLE/ABSTRACT], “social decision-making” [TITLE/ABSTRACT]. Studies were excluded if they were review articles, case-studies, not written in English, primariliy focused on task validation or intended to establish neural/electrophysical correlates. A Prefered Reporting Items for Systemic Reviews and Meta-Analyses (PRISMA) diagram prividing details of the sumber of studies which met the search criteria is included in figure 2."
Figure 2 is attached to in a separate document and the figure legend is below:
"Figure 2. Search strategy: flowchart showing the number of studies included in the systemic review."

Reviewer 2 Report
In the current review the authors summarize the recent results and publication focus on the effects of social cognition, decision making and social isolation in HD.
Althought a broad range of publication have been focus on the cognitive deficits in HD, there is not much data and awareness of the social (sometimes self) isolation that patients with HD suffer. This is crucial for making their social environment understand their behavior even at early stage when other symptoms are not develope.
The review explains in a clear way diferent social problems that the patients suffer an point out that more efforce should be done to raise awareness of these problems.
As a minor point, authors should review the usage of abbreviation. Several abbreviation are being used without previous description and other abbreviations are sometimes used as a full word and sometimes as abbreviation.
Author Response
Point 1: As a minor point, authors should review the usage of abbreviation. Several abbreviation are being used without previous description and other abbreviations are sometimes used as a full word and sometimes as abbreviation.
Response 1: We would like to thank the reviewer for their positive comments and for the observation about the error with the abbreviations. We have read the manuscript and address this point throughout.
Round 2
Reviewer 1 Report
The second version of the manuscript addresses those issues considered problematic in the first review and could be published in its current form.